# Designing ‘Healthy’ Prisons for Women: Incorporating Trauma-Informed Care and Practice (TICP) into Prison Planning and Design

**DOI:** 10.3390/ijerph16203818

**Published:** 2019-10-10

**Authors:** Yvonne Jewkes, Melanie Jordan, Serena Wright, Gillian Bendelow

**Affiliations:** 1Department of Social and Policy Sciences, University of Bath, Bath BA2 7JP, UK; 2School of Sociology and Social Policy, University of Nottingham, Nottingham NG7 2RD, UK; 3Department of Law and Criminology, Royal Holloway, University of London, Egham TW20 0EX, UK; 4School of Applied Social Science, University of Brighton, Falmer BN1 9PH, UK

**Keywords:** wellbeing, mental health in prisons, women’s health, trauma, Trauma Informed Care and Practice

## Abstract

There has been growing acknowledgment among scholars, prison staff and policy-makers that gender-informed thinking should feed into penal policy but must be implemented *holistically* if gains are to be made in reducing trauma, saving lives, ensuring emotional wellbeing and promoting desistance from crime. This means that not only healthcare services and psychology programmes must be sensitive to individuals’ trauma histories but that the architecture and design of prisons should also be sympathetic, facilitating and encouraging trauma-informed and trauma-sensitive practices within. This article problematises the Trauma-Informed Care and Practice (TICP) initiatives recently rolled out across the female prison estate, arguing that attempts to introduce trauma-sensitive services in establishments that are replete with hostile architecture, overt security paraphernalia, and dilapidated fixtures and fittings is futile. Using examples from healthcare and custodial settings, the article puts forward suggestions for prison commissioners, planners and architects which we believe will have novel implications for prison planning and penal practice in the UK and beyond.

## 1. Introduction

Although the relationship between health/emotional wellbeing and low social status has long been at the centre of conceptual thought and empirical analysis, the poor health of prisoners relative to the general population remains under-researched. Social scientists have emphasised the coercive elements of psychiatry and mental healthcare (e.g., [1,2]), which result in vulnerable individuals, including those with severe and enduring mental illnesses, being held in custody rather than recipients of quality, care-orientated, community-based mental healthcare. Female prisoners are of particular concern, as they disproportionately experience physical and psychological problems, exacerbated by substance misuse and mental and physical (including sexual) abuse from childhood [3]. Women in prison report an acutely more painful experience than their male counterparts, with many suffering complex emotional biographies and histories of community-based trauma and abuse pre-imprisonment [4,5]. In England, 65% of imprisoned women have been diagnosed with depression compared to 37% of incarcerated men, and women account for almost a quarter (23%) of all prison self-harm incidents, even though they make up just 5% of the overall prison population [6]. Bloom et al. [7] (p. 31) conclude that ‘addressing the realities of women’s lives through gender-responsive policy and programs is fundamental to improved outcomes at all criminal justice phases’.

With this in mind, we explore a relatively embryonic principle in prison healthcare policy and operational culture: Trauma-Informed Care and Practice (TICP). Following a lecture tour of the UK penal estate in 2017 by Stephanie Covington, a US-based clinical psychologist and pioneer of TICP, the *One Small Thing* ‘trauma-informed’ training package (supported and facilitated by prison philanthropist, Lady Edwina Grosvenor) was rolled-out to staff in women’s prisons across England and Wales. However, during our research, several senior prisons personnel and academic colleagues who conduct research in women’s prisons described TICP as ‘fashionable’ or ‘faddish’; a well-intentioned new initiative that, in the context of the reality of imprisonment, can never be more than ‘window-dressing’. It is beyond the scope of this article to offer any sort of evaluation of the merits of TICP, although we will refer to a recent MQPL (Measuring the Quality of Prison Life) report on the quality of prison life at one women’s prison in England [8] in examining its limitations. Our belief is that, if properly implemented, TICP has the potential to ameliorate the traumas experienced by women in custody, but that it must be viewed as a *holistic* set of practices that include and are inextricably linked to the environmental context in which they take place. Attempts to ‘normalise’ prison interiors as a strategy that goes hand-in-hand with TICP are limited, because many senior prison personnel have narrow ideas about what constitutes ‘normal’. On prison governors’ social media sites and in their public presentations (e.g., at conferences), ‘before’ and ‘after’ photographs of prison cells, association rooms, classrooms and exercise yards, are produced as ‘evidence’ of an enlightened approach. Typically, they illustrate cosmetic improvements in colour palettes and choice of furniture, but still the spaces shown could never be mistaken for anything other than a prison, and many much-vaunted ‘improvements’ are highly superficial.

Put bluntly, then, while there can be little argument that custodial spaces are brightened by a lick of paint, it is nonetheless the case that long corridors, right-angled pathways with poor sightlines, metal staircases, hard surfaces, bars on windows, clanging doors, jangling keys, a performative, macho officer culture, and all the other aesthetic and aural cues associated with confinement, do not mitigate against the ‘abnormality’ of being deprived of one’s liberty and confined against one’s will in an institution. Moreover, all the above arguably exacerbate pre-existing trauma and/or trigger new feelings of distress by recreating elements of, or emotions associated with, the abusive environment that many women offenders have experienced in their pre-prison lives. Given this context, the article considers the female prison estate and its feasibility for a trauma-informed approach to the built environment. Drawing on site visits in four jurisdictions—England and Wales, Scotland, Northern Ireland (NI), and the Republic of Ireland (ROI)—which included lengthy discussions with prisoners, officers, managers and senior prison service personnel, we discuss the relationship between built environment and (the alleviation of) trauma. In short, our aim is to take the sociology of health and illness into new territory, arguing that a women-centred, trauma-informed approach to health and emotional wellbeing in prisons must start with the processes of prison commissioning, planning and design.

First, though, and by way of explanatory context, we provide a snapshot of the current prison crisis and the topography of the female prison estates across the UK and Ireland, followed by a summary of the kinds of traumatic experiences common among women in prison.

## 2. The ‘Prison Crisis’ as it Affects Women in Custody

UK-based criminologists have been referring to the ‘prison crisis’ for at least five decades, yet the problems have considerably worsened since government-imposed austerity measures were introduced in 2010, since when prison budgets have been cut by approximately a quarter and frontline prison staff reduced by one-third, despite prisoner numbers remaining relatively stable. Prisons that accommodate women vary enormously in size, age, and design, which makes it difficult to extrapolate a single variable (e.g., architecture) from the broader picture of the harms inflicted by incarceration in order to claim causal effect. However, in light of our research and the growing literature on the architectural pains of imprisonment from criminology, carceral geography and environmental psychology, we suggest that, in addition to a woman’s ‘imported’ characteristics (age, sexuality, socio-economic group, histories of trauma, offence type, sentence length, motherhood, sexual partnership, etc.), material and architectural ‘deprivations’ have significant impact on the lived experience of imprisonment and the likelihood of negative health consequences. As Howard [9] (p. 69) observes, ‘The quality of our physical environment can be health giving or health destroying’. Therefore, while staffing levels, security factors, education and employment opportunities are important determinants of a facility’s culture and climate, and high-quality healthcare and psychology provision are crucial for prisoners’ chances of recovery from short- and long-term health problems, equally significant to the shaping of a (relatively) healthy or unhealthy carceral experience are a building’s age, architecture, location, interior design and exterior landscaping. These environmental factors will, in turn, determine the suitability of exercise and recreation facilities, whether a women’s prison is located on the same site as a male establishment, whether it has sole- or multiple-occupancy cells, whether it is has provision for mothers with babies, and whether it is operating at or above its certified normal accommodation capacity.

The combined impact of poor environmental conditions, overcrowding and cell sharing, fewer experienced staff, and less ongoing investment in prisons have exacerbated the mental health problems with which many offenders enter custody, resulting in ‘trauma’ becoming the *leitmotif* of imprisonment, even if not formally recognised or termed as such until very recently. Trauma has, in part, manifested itself in dramatically rising numbers of violent incidents and self-inflicted deaths in custody and the prisons inspectorates in all four jurisdictions have highlighted the pressing problems facing the penal systems. Peter Clarke, Her Majesty’s Chief Inspector of Prisons in England and Wales, summed-up the crisis when he said that the ‘simple and unpalatable truth’ about prisons is that they have become ‘unacceptably violent and dangerous places’ characterised by poor mental health, drug use, and the ‘perennial problems of overcrowding, poor physical environments in ageing prisons, and inadequate staffing’ [10] (pp. 8–9).

### 2.1. England (and Wales)

While the crisis has beset every corner of the prison estate, for the approximately 3800 women in custody in England and Wales, the problems are acute. In 2007, Baroness Corston [11] reported on the vulnerabilities of women in prison, most of whom are accommodated in prisons that were not designed for females, far less for the complex and multifarious traumatic histories that women bring into custody with them. In fact, historically, very little planning has gone into the female estate and women have—because of their relatively small numbers in the context of the prison population as a whole—all too frequently been treated as an afterthought; an addendum to the adult male estate. Limitations of space preclude a detailed description of all England’s women’s prisons (for, actually, there are none in Wales—the nearest is Eastwood Park in Gloucestershire) but, being the largest of the four estates with twelve establishments in total, prisons for women in England differ markedly from one another. Consequently, many women in custody are held in facilities designed for men, young offenders, or for some altogether different purpose than imprisonment, with the result that the architecture and environment may inhibit their recovery and rehabilitation and aggravate feelings of depression and/or anxiety. Corston argued for the decarceration of the majority of women in prison, together with a greater use of ‘women’s centres’; community-based services specifically designed as a women-only ‘safe space’ (for offenders and non-offenders alike), which adopt an empowering, holistic, and women-centred ‘one-stop shop’ model, offering multiple services at one premises (e.g., housing, debt advice, counselling, probation and health services), and often also provide access to childcare [12]. This argument is based on the recognition that prison is experienced as ‘both disproportionate and inappropriate’ for most of the women consigned to it and has a ‘catastrophic’ effect on their children [11] (p. i). More recently, Lord Farmer’s recent review on supporting desistance among women and reducing intergeneration crime [13] notes that maintaining healthy, supportive relations with family and other contacts outside prison are ‘utterly indispensable’ (p. 4) and suggests that being imprisoned many miles from one’s family is likely to exacerbate the harms inflicted by imprisonment.

Despite these expert opinions, political commitment to radically overhaul the female prison sector has experienced ‘stagnation and loss of momentum’ [14], with some progress even being reversed, including the closing of several pre-existing women’s centres. In June 2018, the government announced it had abandoned plans to build five ‘community prisons’ for women in favour of new—and loosely-defined—‘rehabilitation centres’. While much of the rhetoric focused on a more liberal approach to punishing female offenders, the greater drive was saving money and one of the outcomes of the government’s reversal of its initial plans to build new, ‘fit-for-purpose’ custodial facilities is that many women in England and Wales are still held in very poor conditions.

### 2.2. Scotland

In Scotland, half of the 400 women held in custody (5.2% of the overall Scottish prison population) are in accommodation designed for men. The majority have historically been accommodated at HMP/YOI Cornton Vale, a national facility for female offenders located in Stirling which, since 2016, has been undergoing phased, permanent closure (with the eventual full shut-down scheduled for 2020). Additional female places are provided on four mixed-gender sites.

With an operational capacity of 119 women, Cornton Vale had a chronic overcrowding problem; in 2012, the prison held 374 women, but it has been as high as 400. When it opened it was viewed as a pioneering facility in architectural terms, because it was designed on a ‘new generation’ campus model (as opposed to a traditional radial template). However, the individual housing units were too small, resulting in prisoners having no respite from housemates, some of whom were volatile, bullying and/or severely traumatised. Additional problems were unwittingly designed in at ‘The Vale’ and exacerbated by inconsistent staff presence. Toilets and showers were constructed outside living spaces with no direct access, resulting in women being told to use the sinks as toilets during the night. Moreover, both toilets and showers were visible at all times to staff and other prisoners, with ‘stable doors’ (usually only concealing one-third of the body of the occupant), being retained in Cornton Vale (and many other prisons) long after being abolished in mental health hospitals.

Dubbed the ’Vale of Tears’ and the ‘Vale of Death’ by the Scottish press, due to its high rate of suicide, the demolition of Cornton Vale, which began in Summer 2017, was welcomed by Scottish First Minister Nicola Sturgeon, who said ‘Scotland’s only women’s prison has been a toxic hot mess for decades. Its destruction is welcome and long overdue’ [15]. A new national facility for women had been planned to replace Cornton Vale, but following an Inquiry into women’s imprisonment, headed by Dame Elish Angiolini QC, the Scottish Justice Secretary Michael Matheson made the following announcement:
I’ve decided that the current plans for a prison for women in Inverclyde should not go ahead. It does not fit with my vision of how a modern and progressive country should be addressing female offending. We need to be bolder and take a more radical and ambitious approach in Scotland [16].
Inverclyde was mired in controversy because it was to be a large facility (300+ beds, with all the attendant problems associated with location and visiting). Nonetheless, the plans were ground-breaking in design terms and it would have been the first prison in the UK and Ireland designed to be ‘gender-responsive’ and ‘trauma-informed’ (see further discussion below). Now, a smaller, 80-bed national prison is being planned for the site on which Cornton Vale stood, together with five regional Community Custody Units (CCUs) throughout Scotland, each housing up to 20 women. The relatively radical approach taken by the Scottish Prison Service is to be welcomed; however, with the second-highest female prison population per capita in Europe (which doubled between 2002 and 2012), we would argue that the Scottish Prison Service’s approach towards women’s imprisonment cannot afford to be anything *less* than revolutionary.

### 2.3. Northern Ireland and the Republic of Ireland

In Northern Ireland, where women make up just 3% of the prison population, figures from March 2018 show 62 adult women (42 sentenced and 19 unsentenced) and six female young offenders (two sentenced, four unsentenced) residing on the Ash House Unit at HMP/YOI Hydebank Wood (a facility designed for and holding predominantly young male prisoners, opened in 2004) [17]. The women’s unit has prompted concerns about ‘excessive’ strip-searching, which no women are exempt from (including pregnant women and girls under 18), the ‘overly restrictive’ security protocols, poor provision of purposeful activity and educational opportunities, the quality of accommodation in which the women are held, and NIPS’s failure to learn lessons from recent deaths in custody [18]. One of the more positive aspects of the facility is Murray House, a six-bedroom unit for women nearing the end of their sentence, who are substance-free, require little supervision, and are trusted to work in the community. The house is domestic in feel, with ‘normal’ living and dining rooms, comfortable furniture, a well-equipped kitchen, and separate bedrooms. Murray House is set in beautiful grounds outside the prison’s secure perimeter. Regarding Hydebank Wood as a whole, the Inspectorate has concluded that it is ‘wrong to run a female prison at the margins of an overwhelmingly male establishment’ and that the mixed population results in a regime for women that is ‘fundamentally disrespectful’ [18] (p. vi). This view is echoed by Moore and Scraton [19] (p. 179) who describe the ‘persistent harassment’ and verbal sexual abuse experienced by the women from the young men incarcerated alongside them, yet which remained unchallenged by the prison staff.

Finally, in the Republic of Ireland, the majority of the 150 female prisoners are held at the Dóchas Centre, a purpose-built facility for women in Dublin on the same site as Mountjoy prison. Constructed in 1999, and considered pioneering at that time, Dóchas was originally a seven-houseblock establishment for the accommodation of 85 women. In 2012, the facility was expanded to house a further 20 women. However, like Cornton Vale, the Dóchas Centre’s ‘single greatest problem’ has been its consistency in operating ‘way in excess of its maximum capacity’ [20] (p. 9). This has had clear knock-on effects, for example, in terms of the ability of the prison to maintain safe and respectful regimes and provide adequate levels of meaningful activity for the women in its care. The issue of overcrowding had clearly not been resolved in the intervening years between that report and our most recent visit to Dóchas, where, due to unmanageable committals to prison from the courts, overcapacity continued to be a significant limiting factor in terms of what the women at Dóchas could access in education, training and purposeful activity.

The only other facility for women in ROI is the female-only wing at Limerick, predominantly a men’s prison built in 1815, which makes it the oldest operational prison in Europe. While women do not come into contact with the men housed at Limerick, the accommodation there is poor, with all prisoners held in catacomb-like cells, with tiny, heavily barred windows and poor ventilation. Access to natural daylight is scarce and the exercise yards are small concrete spaces with high walls topped with razor wire. The environment and accommodation in the women’s section of HMP Limerick were described by the Irish Inspector of Prisons as ‘deplorable’ [21]. The Irish Prison Service (IPS) is currently building a new prison for women in Limerick, taking the accommodation from 24 cells to 42, which will ease the pressure on Dóchas, by providing more prison places for women from the west of Ireland closer to home. During the planning and design process of the new women’s prison, one of the authors (Jewkes) was engaged as a consultant, along with a senior project advisor from the Scottish Prison Service who had previously been involved in the Inverclyde design. They were both able to offer insight into how Limerick’s design could incorporate trauma-sensitive spaces and aesthetics.

However, what is Trauma-Informed Care and Practice (TICP)? To what extent does it inform practice? Additionally, how might trauma-informed expertise be broadened to encompass prison planning and design? The remainder of this article will seek to address these questions.

## 3. Understanding Trauma

While the intention of this article is not to engage in a deep critique of the concept of ‘trauma’, it would be remiss not to briefly address the conceptual conflict surrounding the term, particularly given the primacy awarded to psychiatric definitions. In arguing for a ‘radical understanding’ of ‘trauma’, Burstow [22] (p. 1304) conceptualizes this broadly as ‘a concrete physical, cognitive, affective, and spiritual response by individuals and communities to events and situations that are objectively traumatising’. She further argues that ‘being traumatised’ is not a binary state, simply defined by its traditionally acknowledged ‘symptoms’ such as numbness, disconnection and dissociation. Rather, it is a fluid phenomenon existing on a ‘complex continuum’ (p. 1302), which is less about discrete and easily identifiable symptoms, and more concerned with recognising that trauma leads to pervasive feelings of being ‘overwhelmed’, ‘existentially unsafe’, and finding the world ‘profoundly and imminently dangerous’ [22] (pp. 1302–1303). This is a departure from more clinical, deficiency-based understandings—e.g., as defined by the Diagnostic and Statistical Manual of Mental Disorders (DSM) published by the American Psychiatric Association—which has historically provided a narrower definition of trauma as an event ‘outside the range of usual human experience [that] would be markedly distressing to almost anyone’ (see DSM III). While the most recent DSM (May 2013) has made significant changes, including explicit recognition of sexual violence as a ‘traumatic event’, and extended the ‘exposure’ criteria to ‘vicarious trauma’ (‘repeated or extreme exposure to details of [a traumatic] event’, DSM V), the shortcomings of psychiatric conceptualisations of trauma continue to present problems when diagnosing the roots and triggers of disorders. This is particularly so in terms of limitations regarding ‘survivors of unremitting and recurrent abuse’ [23] (p. 268) and the failure to acknowledge the role of gender and ‘systematic oppression’ [22] in shaping such experiences. Both omissions are central to the life histories of women in prison.

Unsurprisingly, given that many women in prison have high levels of mental health needs (including major depressive disorder, bipolar disorder, schizophrenia spectrum disorder, and schizoaffective disorder) [24], and that healthcare services within prisons are frequently inadequate [8], many prisoners ‘self-medicate’ with alcohol, drugs, and other substances. A further problem is that clinicians themselves are engaged in ongoing disputes, e.g., as to whether borderline personality disorder is treatable or not [25,26] and whether ‘trauma’ can be used to describe one-off emergencies, as well as long-term chronic conditions [8]. A recent survey in England and Wales identified disproportionately more women than men in prison disclosing a drugs problem (41% against 25%) or alcohol abuse (30% against 16%) on arrival into prison [27]. Her Majesty’s Inspectorate of Prisons in England and Wales identified that 41% of women surveyed in 2017 reported mental health difficulties compared with 29% at the previous inspection, and ‘significantly’ more women reported arriving at prison ‘feeling depressed or suicidal’ than in the previous year [27] (p. 55). These factors can, in turn, create an intense and debilitating work environment for staff, with reported ‘burn-out’ and high levels of absenteeism and early departures from the profession (indeed, while beyond the scope of this article, trauma experienced by prison staff is a topic that merits urgent investigation and policy implementation), who themselves might benefit from a more TICP-centred environment; for example, data from the US has shown a 62% decrease in prisoner-on-staff violence following the implementation of a trauma-informed regime at Massachusetts Correctional Institution-Framlingham [28].

For our purposes in this article, ‘trauma’ is utilised in a broad sense to incorporate both individual and collective reactions to discrete and repeat events. This is not to deny the conceptual conflict surrounding the term, but to offer a working definition to facilitate a discussion of how the lived experience of imprisonment interacts with women’s pre-prison lives, which may have been scarred by multifarious forms of trauma. In addition to mental, physical and sexual abuse, these may include loss and bereavement, witnessing parental abuse, being separated from children and other dependents, and—especially among women serving very long sentences—by their offences, whereby feelings of guilt, regret, anger and grief can manifest themselves in forms of inward-facing violence (i.e., self-harm and suicidal ideation), aimed at punishing the self [4]. Moreover, owing to the comparatively small female prison population and commensurately fewer custodial facilities, women tend to be held much further from their homes than their male counterparts, with adverse implications for mental health.

Trauma exposure is frequently identified by women as instrumental in their ‘pathway to crime’, and incarcerated females have often been victims of much more serious offences (e.g., rape and/or grievous bodily harm) than those for which they are convicted (predominantly non-violent drugs and property offences). A total of 57% of women in prison report having been victims of domestic violence and 53% report having experienced emotional, physical, or sexual abuse as a child—though these are likely to be under-reported [3]. Reasons for non-disclosure are complex, but two explanations are that women fear the consequences of reporting offending behaviour of their abusive partners, and that they frequently encounter a culture of disbelief in the criminal justice system about the violence and abuse to which they have been exposed. They are also frequently trapped in a vicious cycle of offending and victimization—victims of controlling behaviour from a partner who may coerce them into offending and/or victims of poverty and neglect which they may in turn pass onto their children.

A growing awareness of the need for TICP-led service delivery has developed amid concerns regarding the complexities associated with treating women with ‘dual diagnoses’ of addiction and mental health disorders, and who also frequently disclose co-occurring and co-morbid experiences of interpersonal trauma—specifically physical and sexual violence and abuse—across the lifecourse [29]. Within Australian mental health services, work to reduce possibilities for re-traumatisation is underway and this involves ‘recognition of lived experience of trauma and the particular “triggers” that may lead to re-traumatisation and re-victimisation’ [30] (p. 2). According to Muskett [31] (p. 52), the key principles of trauma-informed care are: (i) clients need to feel connected, valued, informed, and hopeful of recovery; (ii) the connection between childhood trauma and adult psychopathology is known and understood by all staff; and (iii) staff work in mindful and empowering ways with individuals, family and friends, and other social services agencies, to promote and protect the autonomy of that individual. The findings of Auty et al.’s 2018 MQPL study of HMP Drake Hall indicate that, while these principles are understood, they are not necessarily consistently practiced [8]. Here, the researchers found that examples of trauma-informed care were visible across the prison; however, the aims of trauma-informed practice were not universally understood and some staff and prisoners were disheartened by the distance between the somewhat hyped status of a trauma-sensitive prison and the reality of day-to-day experience. Similar contradictions were found in the fact that many officers were willing to engage with women on an emotional level and as people with complex histories, but did not always situate the behaviour of the women in the context of their past biographical experiences. Moreover, some staff engaged in behaviour that was antithetical to a trauma-informed environment. The forceful removal of clothing from women suspected of hiding contraband, and hospital escorts accompanied solely by male staff are among the examples offered by Auty et al. [8].

It is our contention that some of the triggers of trauma are environmental and that solutions should be sought in design practice as well as operational culture and healthcare delivery. In 2017, the Governor of HMP Drake Hall gave a conference presentation in which he highlighted some of the improvements made to the interiors of the prison buildings as part of their TICP strategy [32]. However, the limitations of trying to introduce trauma-informed, gender-responsive design cues into an environment originally constructed to house World War II munitions workers (and subsequently male prisoners) were plain to see. Put simply, TICP has to start ‘from the ground up’; otherwise, well-intentioned practices may be destined to fail from the outset.

## 4. Trauma-informed Prison Design: Building Emotional Wellbeing into the Built Environment 

Elliott et al [33] note that trauma-informed services should strive to create an atmosphere ‘respectful of survivors’ need for safety, respect, and acceptance’ (p. 467). Key to this, they say, is a ‘welcoming environment’, which includes sufficient personal space for comfort and privacy, absence of exposure to violent/sexual material, and sufficient staffing to monitor behaviour of others ‘that may be perceived as intrusive or harassing’ (ibid). Bateman et al. [30] (p. 4) further note that TICP settings ‘must focus first and foremost on an individual’s physical and psychological safety, including responding appropriately to suicidality’. However, far from being welcoming places that promote feelings of safety and wellbeing, most prisons are fear-inducing environments for many prisoners (and also, for some prison staff and researchers). They are also antithetical to building a sense of autonomy and empowerment. Reception areas, where prisoners are processed on admission into custody, can be particularly damaging because the administrative demands of efficiency (plus procedural and peripheral security) tend to be incompatible with the concerns of the individual prisoner who, when she most needs it, is given little opportunity to discuss the reality of the world she is entering or her fears concerning unresolved problems on the outside (e.g., women who go to court may not expect to receive a custodial sentence and may have made no provision for their children to be cared for in their absence). Clearly, situations such as this are traumatic for the newly arrived prisoner. These opportunities might come eventually, but at the point of greatest stress, the needs of the system come before the needs of the individual. Withstanding ‘entry shock’ is, then, the first of many psychological assaults which the new prisoner has to face, and attempts at suicide and self-harm, the onset of self-destructive psychiatric disorders are most prevalent in the initial phase of confinement [34].

For those who arrive at prison already affected by their negative life experiences, further trauma exposure appears almost inevitable:
Prisons are challenging settings for trauma-informed care. Prisons are designed to house perpetrators, not victims. Inmates arrive shackled and are crammed into overcrowded housing units; lights are on all night, loud speakers blare without warning and privacy is severely limited. Security staff is focused on maintaining order and must assume each inmate is potentially violent. The correctional environment is full of unavoidable triggers, such as pat downs and strip searches, frequent discipline from authority figures, and restricted movement … This is likely to increase trauma-related behaviors and symptoms that can be difficult for prison staff to manage [35] (p. 1).

Furthermore, living in close proximity to others not of one’s choosing can cause significant stress. Crewe et al. [4] (p. 16) highlight the lack of privacy within custodial settings, not only inhibiting prisoners when using the toilet, dressing, washing, etc., but also creating a ‘suffocating’ emotional intensity. As mentioned previously, the ‘effects’ of the built environment are not easy to extrapolate from other intersecting factors that might impinge on an individual’s mental health and wellbeing, but prison receptions are usually profoundly de-personalising in layout and design, as well as in the manner in which they invasively process people. In addition to the perpetrator- (not victim-) orientated dimensions such as intrusive search techniques highlighted by Miller and Najavits, we might add many others, including: harsh, unnatural lighting, sterile spaces (literally and metaphorically), desolate holding cells, loud, unexpected noises, personal possessions boxed up into containers, institutional showers in full view of reception staff, sounds of distress from other inmates, and the fear of not knowing what happens next.

Yet if we pay closer attention to what is known about individuals who have undergone some kind of trauma or distress, with a view to trying to design environments which do not inflict further psychological damage, there are some perhaps fairly obvious design cues that could be incorporated into custodial facilities. Research on other institutional settings is useful here. In his classic 1984 work, *Institutional Settings: An Environmental Design Approach*, Mayer Spivack [36] offers strategies for diagnosing a sick building that is making sick individuals worse. While his comments relate to a wide variety of non-prison institutions, some of the environmental ‘negatives’ identified are highly pertinent to women’s prisons and could help us move towards a trauma-informed model of custodial design. They include the following (adapted very slightly for our purposes): Disorientation (‘Where am I?’); Loss of familiar personal contacts (‘I am abandoned, lonely, ostracized’): Reduction of behaviour repertoire (‘None of the things I usually like to do can be done here’); Loss of territory (‘I have no place to call my own here’); Loss of territorial markers and property props (‘There is no way of letting people know what’s mine, nothing is safe, no place is sacred’); Loss of home route (‘There is no place to go here that I care about’) [36] (pp. 182–183).

Thinking about environmental ‘negatives’ can lead us to think about behavioural ‘positives’ which thoughtful, trauma-sensitive prison design might nurture (these are more heavily adapted from Spivack’s typology). For example: Does removal from particular areas of the prison isolate sources of trouble that triggered, exacerbated or contributed to symptoms of trauma? Can elements of the Indoor Environmental Quality (IEQ) be controlled to reduce undesired stimuli to more tolerable filterable levels?; Can the environment be designed to reduce feelings of incompetence and inability to cope?; Can the physical (and social) environment be designed to induce or support positive redefinition of self and identity?: Does loss of territory eliminate or increase the need to defend oneself (body space, personal space, physical territory or ‘turf’?); Do the same individuals observed in the less overtly carceral spaces of a prison (workshops, education, art classes, gym) seem less hostile, defensive, paranoid, traumatized, etc., than when seen in, for example, the houseblocks? 

More broadly, our SHI-funded research of women’s prisons supported findings of an earlier study of men’s and mixed-gender facilities [37], which found common basic environmental elements that are near-universally desired by people in prison. These were not expressed as mere preferences, but were framed as matters of ontological security which, if not present, are apt to trigger mental instability and trauma. They include: a need for privacy; for socialization; for warmth when it is cold and for effective ventilation when it is hot; for some freedom of movement outside as well as inside; for regular, high-quality family visits; for meaningful and appropriately paid work/education/activities (including essential transferable skills, e.g., use of digital technologies); the ability to undertake a pastime or hobby beyond those traditionally permitted within custodial settings; facilities to cook one’s own food (and perhaps for one’s family) at least occasionally; to experience interaction with nature; and, crucially, to have a high degree of choice, autonomy and control over all these fundamental actions [37]. We believe that these ontological dimensions of lived experience in custody, together with the positive design cues inspired by Spivak’s analysis, could usefully be taken into account in the planning, architecture, and design of new prisons for women.

Design innovation may not be straightforward, however. Contemporary prison architecture has hardly moved forward since HMP Holloway opened in 1852, originally to take adult males. When spatial experiments have periodically been tried (as at Cornton Vale), the tragic consequences of their design flaws have ensured that architects and commissioners have fallen back on the tried-and-tested designs of history. One of the other limitations on design innovation is that the professionals who work on prison commissions commonly specialise in custodial, justice or security portfolios and most have previously designed many other prisons. Architects are self-referential in the sense that they tend to be heavily influenced by their previous work [37] and have difficulty envisaging something radically different from what they have been asked to produce before, or what they ‘know’ prisons to look like from experience. The emphasis tends to lie on the perceived need, or symbolic desire, for the security paraphernalia that denotes ‘this is a prison’ and women’s prisons, therefore, tend to look like men’s prisons, despite the very different experiences and needs their occupants bring to custody with them. In another sense, however, prison architects are not at all self-referential. They have rarely spent much time in prisons, are not often closely related to anyone who has served a prison sentence, and/or cannot easily imagine their female relatives or acquaintances ending up in custody. Prisons, therefore, may fail to generate the kind of empathetic engagement between architects and end users that other institutions do (commissions for schools, hospitals, even residential care homes for the elderly all involve an extended network of active consumers who the architect can identify with personally, as well as professionally; see Buse et al.) [38].

An added problem with designing gender-appropriate custodial facilities is that architects who work on prisons for females are overwhelmingly male. In a recent study drawing-on interviews with fourteen lead architects on new prison commissions, only one (a landscape architect) was a woman [37]. Further, of the four consortia who competed for the new Limerick women’s prison contract in 2017–2018 (each of which included at least a dozen core personnel), there was not a single woman among them. Male architects are no more immune from the dominant cultural repertoire that imbues women in prison with negative and overwhelmingly tragic stereotypes than anyone else. It is little wonder, then, that professionals who design prisons neither design empathetically on the basis of shared experiences and vulnerabilities, nor imbue the eventual occupants of their buildings with positive qualities and potential to radically transform their lives. 

On the other hand, conventional, historic stereotypes that emphasise women’s supposed passive, non-agentic natures (‘sad’, ‘fallen’, and ‘abandoned’ were all used by prison staff and managers during our visits to women’s prisons) may result in greater public acceptance—and, therefore, political will—for creating more pleasant and trauma-informed custodial environments. Women are perceived to be (and are) less of a security risk than male prisoners and the capital outlay that might otherwise be spent on elaborate security paraphernalia and peripheral security measures can be directed instead to more expensive and ‘softer’ materials (wood, glass, green landscaping and so on). When women’s prisons contain Mother and Baby Units (MBUs) or other facilities for prisoners’ children, public tolerance for ‘normalised’ environments increases further. Even politicians who purport to view prison reform for men and women as of equal importance, will use the women’s estate to ‘test’ public opinion with regard to more radical and widespread reform [39]. 

The question is, then, could prisons be designed to heal rather than cause further harm and to arrest or even reverse trauma? Could prison architects borrow some of the architectural cues from pioneering healthcare centres, which are explicitly designed to be trauma-sensitive? Could they, for example, embrace some of the design innovations that underpin architecturally ground-breaking, trauma-informed initiatives in healthcare, such as Maggie’s Cancer Care Centres, many of which have been designed by high-profile ‘starchitects’ such as Frank Gehry, Norman Foster, Richard Rogers, and Zaha Hadid? Each ‘Maggie’s’, as they are universally known, are unique and architecturally striking; many are breathtakingly beautiful. They are linked by design that is defined by inarguably positive qualities: light, space, openness, tranquility, intimacy, views, connectedness to nature, and domestic (i.e., homely and non-institutional) in space and feeling—all of which might be assumed to mitigate some of the physical and mental impacts of trauma.

The newest Maggie’s Centre in Oldham has been designed to counter or ease the effects of chemotherapy, offering ‘a little sensory delight in the parts of the building that you touch’ [40]. Here, there are no cold metal or plastic surfaces that many patients with neutropenia find unpleasant to touch, yet are common in conventional hospitals. Instead, sensory delight comes from the beautiful tulipwood from which the building is constructed. Views through glass walls to the Pennines and a birch tree that grows in the heart of the building, ‘walled in wavy glass like a giant Alvar Aalto vase’ [40] are more than mere aesthetic touches. Aware of the difficulties that some traumatised cancer patients have in making direct eye contact with strangers in an open space, especially when talking about their deepest fears, the architects incorporated the tree and the external views to give them something else to look beyond to. A balcony with a deep overhang, shields patients from direct sunlight and cork panels soften the acoustics.

Of course, even in the architecture of healthcare, Maggie’s are unique projects that would be difficult to replicate in more mainstream hospital environments, let alone in custodial settings. However, if the kind of design—and design philosophies—that underpin Maggie’s Centres seem entirely unachievable in a custodial environment (in the UK, at any rate), the Scottish women’s prison that was planned, and ultimately shelved, represented a concerted effort to design a gender-responsive and trauma-informed environment. HMP Inverclyde was to occupy a site previously earmarked for a new men’s prison in Greenock, north west of Glasgow, and the budget was fixed at what it would have been had the men’s facility gone ahead. However, because the design team (who included HLM Architects, Arup and a group of senior personnel from the Scottish Prison Service called the Women Offender Group) were focused on the needs and experiences of women who come to prison, rather than all the usual situational security apparatus deemed necessary to prevent men from trying to escape, the money could be spent on materials that minimised trauma and ‘softened’ the environment.

Inverclyde was designed with curves and undulations, rather than long, straight corridors and flat planes—not only more aesthetically pleasing but ensuring good sightlines and avoiding traumatised individuals being taken by surprise by someone suddenly appearing from around a corner. Surfaces were to be warm to the touch, finishes were domestic in feel, colour schemes were soothing, and every bedroom (for they were not going to be called cells) was equipped with an en-suite bathroom. Bedrooms were designed to look and feel more like a student residence than prison cells, with fully controllable heating and ventilation, rounded corners and junctions on the furniture, and a desk with a computer. Each bedroom had a large, bar-less window (with curtains) and with views overlooking the stunning landscape, not a perimeter wall. Beds were easily converted into sofas, facing a television, and storage (including drawers under the bed) was ample. The rooms were designed to be ligature-free and to conform to national legal requirements and international standards of best practice, but the designers went much further than either, in creating de-institutionalised, trauma-reducing environments that would give their occupants privacy and safety while also nurturing a sense of autonomy, reflection, and empowerment. As one member of the Women Offender Group put it, ‘the rooms are all about calm and wellbeing. They are not lavish but are comfortable and tranquil’. While most rooms were for single occupancy, some were for two people so that women who were struggling with substance misuse or suicidal ideation could be ‘buddied up’ with a fellow prisoner of their choosing. Rooms for mothers with babies were the same as double rooms, with the second bed replaced by a cot.

The prison was to be built on a large footprint, allowing plenty of freedom of movement and the women were to be entrusted to move around the site largely unescorted. Communal areas of the prison, including reception, the visiting centre and association spaces in the accommodation buildings, were designed to have the look and feel of any other kind of civic building—a shopping mall, airport lounge, or even a hotel. In the ‘services’ area, pictorial signposts (a pair of scissors for the hairdresser, a shopping trolley for the supermarket, a book for the classrooms, etc.) reduced feelings of isolation and incompetence for those prisoners with poor literacy or English language skills. A large proportion of the budget was to be spent on attractive landscaping, to include horticulture and animal husbandry. A stand-alone ‘family help hub’ near the main entrance of the prison offered a welcoming and supportive environment for visitors and provided them with access to social services, charities and third-sector organisations who could assist prisoners and their families—very much like Maggie’s Centres do for cancer patients and their families. A dedicated Mother and Family garden combined therapeutic design and planting with areas of activity and play, while a ‘Multi-Faith Contemplative Garden’ provided outdoor space for spiritual and personal reflection.

Despite promising a pioneering, trauma-sensitive and gender-responsive approach to women’s prison design, the Scottish Justice Secretary, Michael Matheson’s announcement that Inverclyde would, after all, not be built, was relatively uncontroversial. Acknowledging arguments from the 2012 Angiolini Commission [41] and its predecessor south of the border, the Corston Review [11], he said “I believe that accommodating female offenders, where appropriate, in smaller units, close to their families, with targeted support to address the underlying issues such as alcohol, drugs, mental health or domestic abuse trauma is the way ahead”. As discussed, (and partly as a result of the plans for Inverclyde), Ireland, too, has recently moved in a different and more enlightened direction by holding a design competition for the new Limerick prison and committing to a significant refurbishment of the Dóchas Centre. During our research, senior prison service personnel in both Scotland and Ireland expressed to us their desire to look towards Scandinavian prisons such as Halden (Norway) and Storstrøm (Denmark)—where innovative prison design aims to ‘normalise’ custodial environments, using architecture to promote community and ‘social rehabilitation’ [42]—rather than England and Wales, for inspiration for their new women’s facilities. (That said, even the award-winning architecture and aesthetically pleasing environments of the most lauded ’exceptional’ Scandinavian prisons may do little to mitigate the pains of imprisonment (see, e.g., [43,44]), supporting our argument that there is more to a ’humane’ custodial establishment (particularly size and staff-prisoner relationships) than its design. And perhaps in time, there may even be lessons to learn from the United States, the original architects of global mass incarceration, who are beginning to commission (for women, at least) lighter, softer and more ‘homey’ university campus-style carceral facilities (e.g., see the KMD Architects-designed Las Colinas Detention and Re-entry Facility for women, San Diego County, California, which opened in 2014).

## 5. Conclusion: Towards a New Design Manifesto for Women’s Prisons

Although women constitute a minority of prisoners across the UK and ROI, their numbers are rising. Being diagnosed with a mental health condition, often acquired alongside a history of untreated trauma, means that women are sometimes perceived as ‘untreatable’ by health professionals, resulting in criminalisation, especially in crisis, becoming more likely [45]. We do not know what the future of women’s imprisonment in England and Wales will look like. Indeed, prison reform advocates are growing increasingly frustrated with the way in which government promises have been reneged on and women’s imprisonment has been consistently overlooked. Despite the recent introduction of some elements of TICP in existing prisons, the government’s commitment to short-term cost savings at the expense of long-term rehabilitation, results in well-intentioned but woefully unambitious attempts to improve environments that were originally purposefully designed with a punitive and retributive aesthetic. Indeed, more than one of the commentators and stakeholders we talked to described efforts to make prisons trauma-sensitive and gender-responsive as akin to ‘putting lipstick on a pig’.

Part of the problem in England and Wales, then, is that, with the new women’s estate planning on (perhaps permanent) hold, Trauma-Informed Care and Practice is being implemented in old custodial facilities that were, with only two exceptions (Sodexo-operated HMP & YOI Bronzefield and HMP Peterborough), designed and built for entirely different purposes, including a nineteenth-century orphanage (Styal), Elizabethan manor house (East Sutton Park, Foston Hall), isolation hospital (Send), residence for munitions workers (Drake Hall), adult male prison (Downview) and male Young Offenders Institution (Cookham Wood). We suggest that, if we *must* have prisons for women (and, as mentioned earlier, a strategic review of sentencing and custody with a view to drastically reducing numbers of women in prison is long overdue), architects might embrace the positive environmental design concepts inspired by our reading of Spivak and our research in women’s prisons in the UK and Ireland. In summary, prison managers and staff must be cognisant of the spaces within a prison most likely to trigger, exacerbate or contribute to symptoms of trauma and consider how they can be designed to feel welcoming and safe. Prison environments should be designed to reduce feelings of incompetence and inability to cope and support positive (re)definitions of self and identity. Personal spaces, especially cells, should be just that—personal, with opportunities to choose and control levels of heating, air circulation, lighting and privacy. Ideally, people in prison would have some autonomy over the colour schemes in their bedrooms and would also be offered pleasant association spaces that nurture pro-social skills, including the ability to cook and eat together. Crucially, there should be as much freedom of movement outdoors, as well as indoors, as possible (and it should be based on an assumption that most women in prison can be trusted to move around a site unescorted; not implemented with a ‘worst-case scenario’ in mind, as is currently the norm). In addition, where possible, prison planners, architects and managers should work in collaboration with end users on new prison designs; i.e., prisoners, prison officers and other staff. In our research, the only jurisdiction where female prisoners were asked their opinion was at Dóchas and Limerick in the Republic of Ireland, though many struggled to identify what they would like to see in the new Limerick prison, beyond rather mundane improvements such as hanging space for clothes and softer lighting. They were hampered by their limited experience of aesthetically pleasing, nurturing environments and (literally) were unable to ‘think outside of the box’. (Poignantly, when we showed the women held in Irish prisons the architects’ renders for HMP Inverclyde, they were astonished that it was a prison and said things like ‘That’s the nicest room I’ve ever seen’ and ‘That’s nicer than my bedroom at home’).

It goes without saying that provision of care for women at risk of substance abuse, self-harm and suicide is core to trauma-informed practice, but one element of design that has a broad evidence base to support its health-giving properties is access to and interaction with nature. Studies of hospital patients have linked even views of nature to faster recovery times, reduced demand for medication and lower levels of frustration and impatience, while studies of prisons have found that landscapes that incorporate trees and the wildlife they attract reduce feelings of sterility in the carceral environment and lead to general improvements in emotional wellbeing (see, e.g., [37,46]). Yet, an Inspectorate report of HMP Foston Hall found that women on the remand wing ‘did not have daily access to the open air’ and one-third of women there ‘had less than four hours out of cell each weekday’ [27] (p. 57). Furthermore, even establishments with attractive landscaping—HMP Hydebank Wood being an example, with its extensive green spaces, attractive flowerbeds, horticulture, polytunnels, goats and chickens—may only permit a privileged few to use them. Meanwhile, we witnessed the most vulnerable women at Hydebank being held in solitary confinement in oppressively unventilated, basic cellular confinement that appeared to offer nothing in terms of positive stimulation.

Our conclusion, then, is that the gender–prison–environment–trauma nexus requires theoretical, empirical, policy and practice attention and must be seen as a confluence of circumstances to be regarded *in toto*. It is not enough for prison staff to speak a trauma-sensitive language, or even engage in trauma-informed practice, if it is not fully embedded in the prison’s culture, fabric and design philosophy. When implemented in unsuitable or even dangerous trauma-generating environments, a trauma-informed mode of engagement may be of no greater value than a disregard for imprisoned women’s complex histories and biographies.

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
