# Peer review of "Designing ‘Healthy’ Prisons for Women: Incorporating Trauma-Informed Care and Practice (TICP) into Prison Planning and Design"

_ijerph, 2019, doi:10.3390/ijerph16203818_

Round 1

Reviewer 1 Report

This is an excellent, well written paper that is easy to ready, novel and relevant to key issues of the day.

The methodology is set out well and the background to the contexts in the 4 jurisdictions studied is succinct and clear. Similarly, the discussion of the complexity of defining 'trauma' is well handled and the working definition settled upon is clear and useful.

 I would make some very minor points for the authors to consider, but they are more stylistic than substantial.

p2: 53 - The term MQPL is introduced but the reader is not told what it stands for. There is a helpful footnote, but it is not clear what the letters stand for.

p5: 181-185 - The authors speak of a hypothetical argument being potentially made by 'sceptics'. Do they endorse this argument? This could be more clearly stated. 

p10: 449 - should 'measured' read 'measures'?

Reviewer 2 Report

I found this manuscript to be well written, interesting, and important to the field of corrections. I have only three comments, which I invite you to consider or not.

First, I feel that it would be good to consider acknowledging that a trauma-informed facility staffed with poorly trained employees will not promote safety, wellness, and healing.  I feel the article places too much emphasis on the building. Poorly designed institutions staffed with compassionate, professional employees can mitigate the unwholesomeness of the environment, but I am not so sure a trauma-informed design can reduce the harmful effects of poorly-trained staff.

Second, San Diego County in the US has a beautifully-designed facility for women.

https://www.citylab.com/design/2015/08/future-jails-may-look-and-function-more-like-colleges/402151/

I think this warrants some attention.

Lastly, you mention Scandanavian facilities. I think your readers would like to learn just a bit more about this. What is it, precisely, about those facilities that make them so humane?
